# A Novel Mastadenovirus from *Nyctalus noctula* Which Represents a Distinct Evolutionary Branch of Viruses from Bats in Europe

**DOI:** 10.3390/v16081207

**Published:** 2024-07-26

**Authors:** Anna S. Speranskaya, Alexander V. Dorokhin, Elena V. Korneenko, Ivan K. Chudinov, Andrei E. Samoilov, Sergei V. Kruskop

**Affiliations:** 1Scientific Research Institute for Systems Biology and Medicine, Federal Service on Consumers’ Rights Protection and Human Well-Being Surveillance, 117246 Moscow, Russia; 2Independent Researcher, Ha-Metsadim 66 Maale-Adumim, 9842066 Jerusalem, Israel; 3Saint Petersburg Pasteur Institute, 14 Ulitsa Mira, 197101 Saint Petersburg, Russia; 4Phystech School of Biological and Medical Physics, Moscow Institute of Physics and Technology, Institutskiy per. 9, 141701 Dolgoprudny, Russia; 5Zoological Museum, Lomonosov Moscow State University, Bolshaya Nikitskaya 2, 125009 Moscow, Russia

**Keywords:** Chiroptera, *Mastadenovirus*, novel species, viral distribution, Vespertilionidae, *Nyctalus noctula*, *Pipistrellus nathusii*

## Abstract

Bats are natural hosts of a wide variety of viruses, including adenoviruses. European bats are known to carry mastadenoviruses categorized as species B (widespread in European Vespertilionidae bats) and whose taxonomy has not been clarified. We examined fecal samples from Vespertilionidae bats (five species) captured in central Russia and found that 2/12 (16%) were positive for mastadenoviruses. The partial genome of the mastadenovirus was assembled from *Pipistrellus nathusii*, representing the bat adenovirus species B. The complete genome (37,915 nt) of a novel mastadenovirus was assembled from *Nyctalus noctula* and named BatAdV/MOW15-Nn19/Quixote. Comparative studies showed significant divergence of the Quixote genome sequence from European bat mastadenoviruses, while the only known virus showing low similarity was the isolate WA3301 from an Australian bat, and together they formed a subclade that separated from other BatAdVs. Phylogenetic and comparative analysis of the protein-coding genes provided evidence that Quixote is related to a novel species within the genus *Mastadenovirus*, provisionally named “K” (as the next available letter for the species). Phylogenetic analyses revealed that some earlier viruses from Western European bats, for which only partial DNA polymerase genes are known, are most likely members of the tentatively named species “K”. Thus, at least two species of mastadenovirus are circulating in bats throughout Europe, from western to eastern areas.

## 1. Introduction

Adenoviruses (double-stranded DNA viruses of the family *Adenoviridae*, with genomes of 25–48 kb) have been intensively studied because of their role as pathogens in humans and domestic animals [1,2,3,4]. In addition to humans, mastadenoviruses (MastAdVs) can infect a variety of mammalian hosts. In general, MastAdVs appear to be host-specific viruses [3], but there are a few exceptions. For example, canine adenoviruses (CAdVs) could infect a wide range of carnivores [5,6,7]. As another example, human adenoviruses (HAdVs) have been detected in monkeys [8,9] and could be transmitted from primates to humans, or vice versa [8,9,10,11,12].

Bats (Chiroptera) are known reservoirs of several mammalian viruses, some of which can cause human disease or are ancestors of several human viruses. Mastadenoviruses have been isolated from chiropterans worldwide. To date, bat mastadenoviruses have been classified into ten species (Bat Mastadenovirus A–J) according to ICTV (International Committee on Taxonomy of Viruses) data [1], forming three clades on the phylogenetic tree, associated with Vespertilionidae, Rhinolophidae or Miniopteridae/Pteropodidae bats [13,14]. Clade 1 includes viruses isolated from Vespertilionidae bats (BatAdVs species A, B, G and J), which are closely related to canine adenoviruses. Clade 2 includes viruses from Rhinolophidae bats (BatAdVs species C) and clade 3 includes viruses from Miniopteridae and Pteropodidae bats (BatAdVs species D, E, F, H and I) [13]. Information on mastadenoviruses in European bats has been published regularly since 2008, with the first surveys conducted in Germany [15]. To date, bat mastadenoviruses have been detected in several European countries, including France [16], Germany [15,17,18], Hungary [18,19], Italy [20], Spain [21], Sweden [22], and Switzerland [23]. However, only one complete genome is available for European isolates: Bat adenovirus 2 strain PPV1 from *P. pipistrellus*, captured in Germany in 2007 [15,17].

The majority of mastadenoviruses from European bats belong to species B or an unidentified taxonomic position, which could be explained by the current incompleteness of the genome sequences. A number of studies that noted the high diversity of mastadenoviruses from bats in Germany, Hungary, and Italy only analyzed short genome fragments. Some of the investigated sequences showed high nt identity with the BatAdVs species B, but the taxonomy positions of some viruses were unclear [18,19,20,23]. Some researchers have clearly suggested that mastadenoviruses other than BatAdVs species B circulate in bats in Europe. For example, a total of 1717 samples collected in Spain between 2004 and 2016 from bats representing 27 of the 32 European bat species revealed multiple BatAdVs. The authors declared that the new groups are different from the previously described BatAdVs species A and B, but without clarification, as they only sequenced partial hexon or DNApol genes [21].New data on the genetic diversity of bat adenoviruses may be helpful in better understanding the evolution of adenoviruses in bats and their distribution across Europe. Here, we report the characterization of mastadenoviruses obtained from fecal samples of Vespertilionidae inhabiting the Central European part of Russia (Zvenigorodsky district, Moscow region), including the genomic characterization of novel mastadenovirus species obtained from *Nyctalus noctula*.

## 2. Materials and Methods

### 2.1. Sample Collection

Bat fecal samples were collected in the summer of 2015 in the Zvenigorodsky district of the Moscow region (Sharapovskoe forestry, coordinates N55.69, E36.70), as described in [24]. Bats were captured and sampled by professionally trained staff from the Biology Department of Lomonosov Moscow State University and released at the site of capture. The collected swabs were placed in a swab transport and storage medium with mucolytic agent (AmpliSens, Moscow, Russia) at 4 °C during transport to the laboratory and then stored at −80 °C prior to processing.Twelve bats of five species, *Myotis dasycneme* (n = 2), *Myotis daubentonii* (n = 2), *Myotis brandtii* (n = 1), *Nyctalus noctula* (n = 2), and *Pipistrellus nathusii* (n = 5), were analyzed.

### 2.2. Nucleic Acid Extraction, Reverse Transcription, Library Preparation and High-Throughput Sequencing

Nucleic acid was extracted using the QIAamp Viral RNA Mini Kit (Qiagen, Hilden, Germany) during our previous investigations of RNA viruses. The 140 µL of resuspended fecal sample was used for a start. Further steps were performed according to the original protocol. Due to the ineffective action of DNAase, the residual DNA remained in the resulting nucleic acid extracts. A total of 10 µL of RNA/DNA mixture was taken for reverse transcription using Reverta-L (AmpliSens, Russia). Second-strand cDNA was obtained using the NEBNext Ultra II Non-Directional RNA Second Strand Module (E6111L, New England Biolabs, Ipswich, MA, USA). To increase the input concentration, 24 µL of first-strand product was added to 10 µL of H_2_O (milliQ, Merck-Millipore, Darmstadt, Germany) for the subsequent steps.

Double-stranded cDNA was used for the library preparation using the NEBNext Ultra II DNA Library Prep Kit for Illumina (New England Biolabs, USA). The end prep was performed according to the manufacturer’s protocol. For the adaptor ligation step, Y-shaped adaptors compatible with the Nextera XT Index Kit were used at 4 pM per reaction. PCR amplification with index adaptors at 7.5 pM per reaction was performed using the Nextera XT Index Kit (Illumina Inc., San Diego, CA, USA) in 25 µL total volume according to the NEBNext Ultra II DNA Library Prep Kit for Illumina protocol with 15 cycles.

High-throughput sequencing was performed on an Illumina HiSeq 1500 (Illumina, USA) using the HiSeq PE Rapid Cluster Kit v2 and HiSeq Rapid SBS Kit v2 (500 cycles) (Illumina, USA).

### 2.3. Genome Sequence Assembly and Annotation of Novel Virus

Paired reads were filtered using Trimmomatic with the parameters SLIDINGWINDOW:4:25 MINLEN:40. Preliminary analysis showed that there were almost no human reads in the metagenomic sequences, so there was no need to filter them out. The *Pipistrellus nathusii* and *Nyctalus noctula* reads were also not filtered out because at a time when all of the bioinformatics analysis was performed, there was none of these species genomes available. After read trimming, the reads were assembled using SPAdes v.3.14 in metaSPAdes mode [25]. Contigs shorter than 500 nt were filtered out and the remaining contigs were aligned to viral sequences in both the nt database using blastn and nr using blastx. Full genomes were selected by comparing the length of the contig with the median length of the top hits (with a threshold of 90% of the maximum bit score) of the blastn alignment to the nt database or, in the case of the blastx alignment, with the median length of the sequence from which the protein in the nr database was derived. Contigs with a length difference of more than 5% from the median were filtered out, and the remaining contigs were considered as full genome candidates and were subsequently checked manually to account for possible errors in the algorithm and the NCBI databases.

Genome annotation was performed using the following software: de novo using Prokka software [26] and reference-based using MK472072.1 as a reference, GATU [27] and and ViPTree software version 4.0 [28]. The ORFs/features were additionally revised by BLAST-based similarity/homology searches. The coding sequence domains were predicted as open reading frames (ORFs) that shared certain features, such as nucleotide/amino acid sequence similarity, with known viral genes in the GenBank database. Viral ITR sequences were determined by BLAST. The viral map was visualized using the SnapGene Viewer (“SnapGene software (www.snapgene.com)”).

### 2.4. Phylogenetic Analysis 

The sequences of the newly obtained virus isolate were aligned with homologous sequences of closely related viruses available in the public database—GenBank (Appendix A). Alignment was performed using the Multiple Sequence Alignment (MSA) tool, MAFFT version 7 [29], using the default parameters. Phylogenetic analyses were performed using W-IQ-TREE (multicore version 2.2.2.3) with ModelFinder [30], tree reconstruction [31], and ultrafast bootstrap (1000 replicates) [32]. The maximum likelihood (ML) phylogenetic tree was constructed. The best-fitting nucleotide substitution model was determined as SYM+I+G4. Bootstrap analysis with 1000 replicates was performed to assess the robustness of the tree topology. The resulting tree topology was visualized and annotated using ITOL version 6.8.1 [33].

### 2.5. Maps Generation

The maps were generated using the Python (v3.9.17) Plotly package (v5.9.0) [34,35]. Briefly, the corresponding sample coordinates were transformed into the latitude and longitude, stored as a table with respect to the sample metadata, and plotted from the dataframe using Plotly Express.

## 3. Results

### 3.1. Mastadenovirus Detection

The sequencing data obtained earlier during the analysis of the distribution of the coronaviruses were used to search for *Adenoviridae*. In short, fecal samples from bats captured in 2015 in the Moscow region were investigated. Two of the twelve samples (16%) were positive for mastadenoviruses: *Nyctalus noctula* (sample №19) and *Pipistrellus nathusii* (sample №21). 

The complete genome of the MastAdV was assembled from *N. noctula* and named BatAdV/MOW15-Nn19/Quixote (short version: Quixote). The final consensus had a 37,915 nt size, the average coverage was 19.4, and the median coverage was ~45 (Appendix A). The consensus of the Quixote sequence has been deposited in GenBank under accession number PP297886.

The partial genome of a MastAdV was obtained from *P. nathusii*. De novo assembly yielded 12 contigs with average read coverage of 2.4–6.0, representing mastadenovirus genome fragments with a total of 7831 nt (24,7% coverage of the reference genome NC_015932.1). The isolate was named BatAdV/MOW15-Pn21 and the genome contigs were deposited in GenBank under accession numbers PP386572–PP386580 and PP997449-PP997451. The assembled contig coordinates relative to the reference genome of the Bat adenovirus 2 (NC_015932.1) are presented in Appendix A. The contig mapping to the reference is presented in Appendix A.

### 3.2. Genome Characterization of Novel Mastadenovirus BatAdV/MOW15-Nn19/Quixote

The genome of Quixote (isolate BatAdV/MOW15-Nn19) contains 35 open reading frames and the typical gene order found in mastadenoviruses (Figure 1). The genome is flanked by inverted terminal repeats (ITRs, origins of viral DNA replication), where the length of the ITR was determined as 59 nt by analysis of the NGS data assembly. Since the ITR sequences at the 5′- and 3′-terminal ends of the genome were identified, we considered the assembled genome to be complete.

All the identified genes/ORFs of the Quixote virus genome have been accurately annotated according to the ICTV description of the MastAdV genome structure [1]. All the genes described as characteristic of mastadenoviruses were identified in the genome of the newly assembled virus (Table 1). In addition, eight ORFs encoding polypeptides of unknown function were found. We compared each of the identified ORFs (by nucleic acid or amino acid sequence) of the Quixote virus with the GenBank and UniProt records and found that most of the identified ORFs (27/34) encode proteins that show significant alignments with the annotated proteins encoded by genes from the mastadenovirus isolate WA3301 from *Chalinolobus gouldii* (Vespertilionidae bat from Australia, genome acc. number MK472072.1), with the BLAST homologue identities ranging from 29.21% to 86.52% (Table 1). We also found that the pre-core protein X encoded by gene *L2_4* is most closely related to that of *Rousettus leschenaultii* (Pteropodidae bat from China, YP_009388318). The predicted protein encoded by the *E3* gene has homologous to the product of the *E3* gene of the guinea pig adenovirus (30.65% identity to YP_010796290). The protein encoded by *E4 orf6/7* showed a similar identity (44.59%) to that of the human mastadenovirus *D* (AGT77890) (Table 1).

The assembled genome of the Quixote virus has a 37,915 nt size, with 49.9% GC content like the closest isolate WA3301 (37,617 nt and 49.1% GC content). At the same time, other BatAdVs species that are close to the new virus have a length range of complete genomes of 31,629–31,806 nt (5000 shorter) and a GC content of 53.5–56.9% (for example, BatAdV-2 (JN252129), BatAdV-3 (GU2269700 and other BatAdVs isolated from the Vespertilionidae bat, see Figure 2). The mastadenovirus genomes range in size from 27,952 (polar bear adenovirus 1) to 37,860 nt (simian adenovirus 31.2 from chimpanzees, species human mastadenovirus C) [1,36,37] and in nucleotide composition (GC) from 43.6% (BAdV-2) to 63.9% GC (PAdV-3) [38]. This means that the genome size of the new adenovirus is within the known range, but the genome size of the new virus is unusual for bat adenoviruses.

The genome organization of the Quixote virus was the same as that of isolate WA3301 in the E3 region. However, Quixote differs from WA3301 in the E4 region, as well as both (Quixote and WA3301) differing from other bat mastadenoviruses in the E4 region. We have highlighted the similarity of the E3 region and the difference in the E4 region precisely because, according to the ICTV criteria [1], these regions are particularly important for species separation in mastadenoviruses. See Table 1 for details.

Four of the nine ICTV species delimitation criteria for adenoviruses could only be used for viruses identified by NGS methods, namely the phylogenetic distance (>10–15%, based on distance matrix analysis of DNA polymerase amino acid sequence), genome organization (characteristically in the E3 region), nucleotide composition, and host range. According to the ICTV criteria, BatAdV/MOW15-Nn19/Quixote is related to a new species of bat mastadenovirus: the full genome sequence showed 72.41% of the identity of nt (query coverage was only 0.53) with the only known closest virus, isolate WA3301 (MT815933.1) isolated from *C. gouldii* captured in Australia in 2018 [39]. For comparison, the homology of two different strains of the same species “B” from bats caught 10 years apart in different European countries is 0.99, with query coverage of 100% (NC_015932.1 vs. MK625182.1). As another example, the homology of representatives of species “B” and species “G” is 0.77, with query coverage of 35% (NC_015932.1 vs. NC_031948.1).

For short sequences, the DNA polymerase amino acid sequence of Quixote showed 38% shared identity (0.27 query coverage) with isolate WA3301 (MK472072.1). The next closest virus was Nynoc/Switzerland/2019 (MT815933.1), which was found in *N. noctula* with 5% shared identity. The new species can be provisionally named “K”, because the last known species of mastadenovirus is named after the previous letter of the alphabet, “J”.

### 3.3. Phylogenetic Analysis of Bat Mastadenoviruses from Russia

The genomic nucleotide sequence of the novel virus was phylogenetically analyzed against the full-length genomes of representative members of the genus *Mastadenovirus*. We found that Quixote (isolate BatAdV/MOW15-Nn19) from *N. noctula* and isolate WA3301 from *C. gouldii* form a subclade that was clearly separated from other Vespertilionidae BatAdVs (Figure 2). Both hosts of the novel viruses, *N. noctula* and *C. gouldii* (Gould’s wattled bat), belong to the bat family Vespertilionidae. Interestingly, both viruses were found in geographical regions as far apart as possible: European Russia and Australia.

For taxonomic identification of the second MastAdV identified in this study—BatAdVs/MOW15-Pn21 from *P. natusii*—a phylogenetic tree was constructed using the partial sequence of the DNA polymerase (DNApol) genes. The sequences of the two obtained viruses, BatAdV/MOW15-Nn19/Quixote and BatAdVs/MOW15-Pn21, were phylogenetically distant from each other. The isolate BatAdV/MOW15-Pn21 was similar to bat adenovirus 2, which is a typical isolate of species B, according to phylogenetic analysis of the 5′-terminal part of DNA polymerase encoding genes (Figure 3a). BLAST investigation results in the closest nt identity (99%) with the BatAdV from Swiss bats, 2019, *P. pipistrellus* (MT815935.1) and nt identity in the range of 79.8–80.1% to BatAdVs from bats captured during 2017–2019 in Switzerland, *P. pipistrellus* (MT815928.1, MT815929.1), Spain, *P. pipistrellus* (MN490088.1), and Italy, *P. kuhlii* (MK625182.1).

According to the phylogenetic tree constructed using the terminal part of the DNApol, the BatAdV/MOW15-Nn19/Quixote falls into a clade with the isolate WA3301 from *C. gouldii* from Australia and isolate Nynoc/Switzerland/2019 from *N. noctula* from Europe.

The additional phylogenetic analysis of the central segment of the DNA polymerase gene was carried out in relation to recently studied viruses (it allows for increasing the sample of viruses analyzed) (Figure 3b). In the context of the additional DNA polymerase gene analysis, the Quixote demonstrated similarity with a wide spectrum of MastAdVs from European bats, including those identified in *N. noctula* Nynoc/Switzerland/2019 (MT815933.10 and a couple of viruses identified in the Spanish bats *Nyctalus lasiopterus* and *Nyctalus leisleri* (JX065118, JX065125, JX065128, JX065124, JX0651270). The Australian virus—isolate WA3301 obtained from *C. gouldii*—falls into a different clade with viruses from *Hipposideros larvatus,* which were captured in China, Yunnan in 2015 (OP963609), and *Phyllostomus discolor* captured in Brazil in 2021. This observation suggests that a novel mastadenovirus from *N. noctula* represents a distinct evolutionary branch of viruses infecting bats in Europe.

All bat species harboring viruses similar to Quixote are Vespertilionidae. Quixote was identified from *N. noctula* and similar Quixote-like viruses were identified from different species of the same genus *Nyctalus* (*N. noctula* captured in Switzerland, *N. lasiopterus* and *N. leisleri* captured in Spain) or other Vespertilionidae (*C. gouldii*, from Australia), see Figure 4.

## 4. Discussion

### 4.1. Prevalence of Mastadenoviruses in Bat Populations

Adenoviruses have been detected in bats across Europe, from Italy to Hungary. Here, we report MastAdVs in two of the twelve examined bats from Russia. Despite the small sample size, we believe it is generally necessary to note the prevalence of MastAdVs in bats, as our data are consistent with previously published studies [15,16,18,19,20,23,40,41]. The estimated range of MastAdVs-positive bats varied from 14.73% (German and Hungarian bats) to 17.4% (Italian bats) [19,20]. Some studies provide a different estimate. For example, an analysis of Swiss bats found adenoviruses in only 2.6% of the animals [23] and 3.6–8.3% of bats from Spain, depending on the type of sample examined, oropharyngeal or fecal [21]. It appears that high percentages of bats carrying adenovirus are the norm in the wild.

BatAdVs have been isolated several times from European bats. However, the number of described species of bats captured in Europe is quite small. In fact, for most BatAdVs from European bats, only fragments of the genome are known. Phylogenetic analyses are usually based on short fragments of 200–450 nt of DNA polymerase or hexon genes or short read sequences (70 nt on average) from the pool of samples; for example, in France [16]. This approach significantly reduces the accuracy of the research and the reliability of the conclusions drawn.

The BatAdV/MOW15-Nn19/Quixote from *N. noctula* is suspected to represent a new species, provisionally designated as species “K” (not approved by the ICTV). This finding is based on both whole-genome analysis and phylogenetic analysis of the DNA polymerase gene. Previously, a phylogenetic analysis of a large number of short MastAdV sequences obtained from the Spanish bats allowed the authors to suggest the existence of new members of the genus *Mastadenovirus*, in addition to the species already described. But the authors sequenced the short fragments of the DNA polymerase and hexon genes, so the existence of these species remains a guess [21]. Here, we found similarity between the Quixote and some isolates from European bats captured in Spain when we analyzed a short (central) segment of the DNA polymerase gene (proposed as the new members of the genus *Mastadenovirus* by [21]). Also, we found similarity between Quixote and the virus from Swiss bats (strain Nynoc/Switzerland/2019). Thus, at least two endemic species of mastadenoviruses circulate in bats in Europe: species B and K.

### 4.2. Genomic Features of the Provisionaly Novel Mastadenovirus Species

For the Quixote genome, the proposed start codons of two genes—*E1A* and *POL* (DNA polymerase)—are Val (instead Met). These results were obtained using the “prokka” annotation tool. The standard mammalian genetic code was used as the translation table for the genus *Mastadenovirus*. Then, the coordinates were manually checked using *.bam files with careful consideration and using the closest complete genome as a reference (accession number MK472072.1). For the *E1A* gene, the start-codon position was covered by 71–72 reads (GTG sequence nucleotide count—99–100%). For the *POL* gene, the start-codon coverage was 70 reads, while the GTG (reverse complement of CAC) sequence nucleotide count was 97–100%. Some findings suggest that valine tRNA may be involved in the initiation of sgRNA translation [42]. Therefore, we suggest possible molecular mechanisms leading to the replacement of Val by Met during translation. But these results show that new viral genomes require experimental validation of the gene coordinates.

### 4.3. Could the Pipistrellus Bats Distribute Mastadenoviruses around the Old World until Australia?

In this paper, we report two different BatAdVs from bats captured in Russia (from Central European part of Russia). According to phylogenetic analysis of the DNA polymerase gene, BatAdV/MOW15-Pn21 from *P. nathusii* is related to species B, as the first BatAdV identified in a European bat (BtAdV-2 isolate PPV1 from *P. pipistrellus*, captured in Germany), as the majority of BatAdV strains identified in Italy in 2019 [20] and Spain [21]. With Russia being the easternmost area of Europe, the results presented here show that members of species B are widespread in bats across Europe. However, a question now arises as to whether the same viruses are present in bats that live on the other side of the Ural Mountains in Siberia.

Mastadenoviruses tend to be host-specific [3,21]. Species B mastadenoviruses have been identified in vespertilionid bats captured in Europe. Representatives of species A have been described in vespertilionid bats from South Asia, namely China and Japan (typical genomes are GU226970 and LC385828), while representatives of species G are found in North American vespertilionid bats (typical genome is NC_031948.1) and species J—in Japanese bats (typical genome is LC385827). The novel BatAdVs presented in this paper fit well with the concept of host specificity. The large distance between the geographical locations where the Quixote-like viruses have been identified raises a question about how these viruses spread across the globe. The closest BatAdV to BatAdV/MOW15-Nn19/Quixote from *N. noctula* (Russia) was found to be WA3301 from *Chalinolobus gouldii*, Australian Vespertilionidae. There are significant zoological barriers between Europe and Australia. *Chalinolobus* (and three other genera of Vespertilionidae) are endemic to Australia. The time of divergence of *Chalinolobus* from other genera is estimated to be around 12 million years [43], presumably the time of their invasion of Australia via a chain of islands including New Guinea. The genus *Chalinolobus* are the “old” Australian vespertilionines, so it is unlikely that they were the first hosts of viruses related to European viruses. We could assume that the MastAdV WA3301, which is related to the European Quixote, was acquired by *Chalinolobus* from some intermediate host. However, another possible route is distribution by bats, which is the most likely bridge for the spread of bat viruses from Europe through Asia to Australia: *Pipistrellus* and *Myotis* bats.

Here are some reasons in favor of this: all European Pipistrellus belong to the “west- ern” clade of the genus and there is extreme phylogenetic proximity between the “west- ern” clade *Pipistrellus* and the genus *Nyctalus* [44,45]; therefore, the presence of related viruses in these two genera could be expected. *Pipistrellus* and *Myotis* bats have a large range, including the Old World and Australia (*Pipistrellus*), or a global distribution (*Myotis*) [46]. Both *Pipistrellus* and *Myotis* are probably able to cross sea straits and colonize islands. This possibility has been documented for the European *P. nathusii* [47] and is also inferred from the fact that some *Pipistrellus* and *Myotis* species have large insular ranges. The Australian *Myotis macrotarsus* has close relatives in Asia: *Myotis horsfieldii* and others. *Pipistrellus* bats currently living in Australia have possibly close relatives in the Sunda Islands and tropical Asia [48]. If intermediate evolutionary forms of mastadenoviruses between European and Australian mastadenoviruses are found in Vespertilionidae bats in central or south-eastern Asia in the future, it could provide evidence for this mode of geographical distribution of bat mastadenoviruses.

## 5. Conclusions

At least two bat MastAdV species are endemic to Europe: the previously described bat adenovirus species B and the novel bat adenovirus species described here (provision- ally named “K”). The complete genome of a new species of the genus *Mastadenovirus* (tentatively named “K”) from bats living in the European part of Russia was determined. Based on the analysis of short fragments of the mastadenovirus polymerase genes identi- fied in European bats, we hypothesized that some of them also belong to the described new species. We also found mastadenovirus species B in the European part of Russia, which is widespread in bats captured in Europe. All of this suggests that closely related bat MastAdVs are circulating in bats throughout Europe (from western to eastern areas).

## Figures and Tables

**Figure 1 viruses-16-01207-f001:**
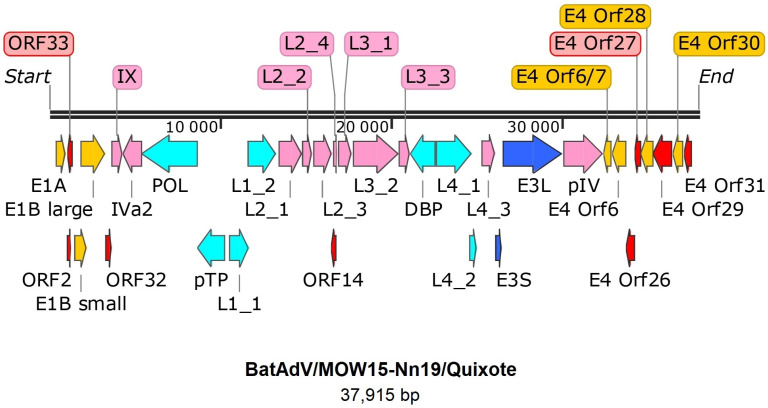
Gene map of BatAdV/MOW15-Nn19. The genes are shown as arrows. Pink—structure proteins of virion, cyan—DNA synthesis and package, dark blue—host interaction (host defense) associated proteins, yellow arrows indicate genes that encode proteins that modulate the host cell’s transcriptional machinery, red arrows—predicted proteins without homologs determined.

**Figure 2 viruses-16-01207-f002:**
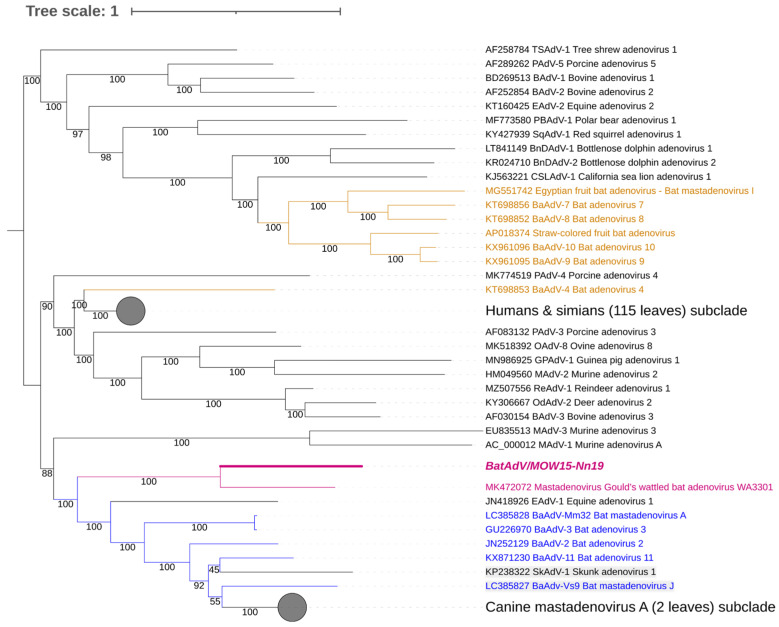
Phylogenetic analysis of mastadenovirus full genomes. The phylogenetic tree was constructed from 153 mastadenovirus genome sequences from GenBank and the novel bat mastadenovirus BaAdV/MOW15-Nn19/Quixote. Numbers in branches indicate bootstrap values. Best-fit substitution model according to BIC: SYM+I+G4. A new mastadenovirus, Quixote, is marked in bold and purple; its closest relative, isolate WA3301, is only purple (both viruses isolated from the Vespertilionidae bat). Other mastadenoviruses also isolated from the Vespertilionidae bat are marked in blue. Mastadenoviruses whose hosts belong to other bat families are highlighted in brown.

**Figure 3 viruses-16-01207-f003:**
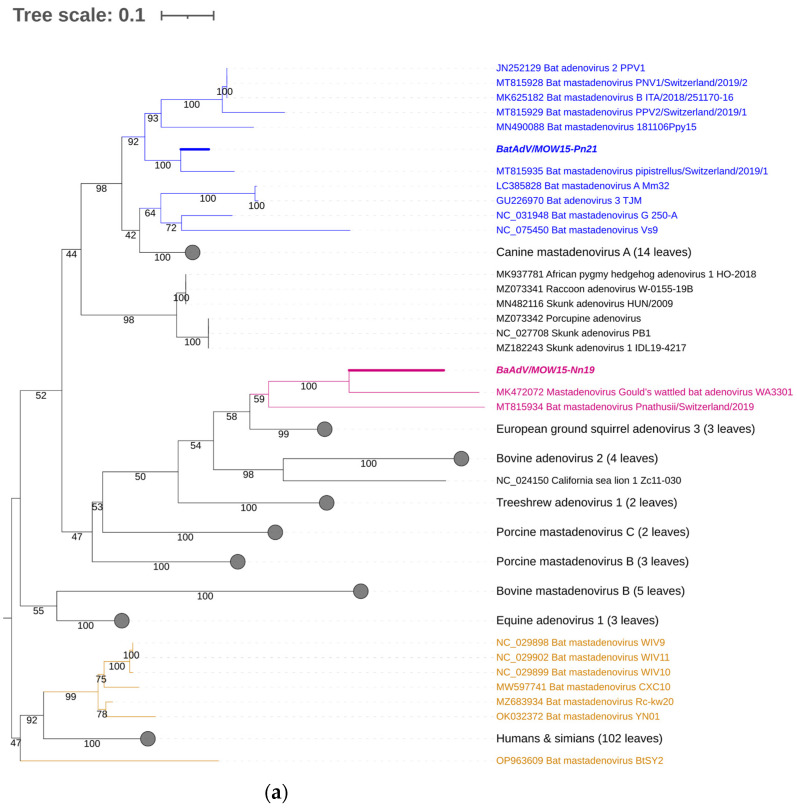
Phylogenetic analysis of the MastAdV DNA polymerase gene. (**a**) The phylogenetic tree constructed using the 5′-terminal part of the DNA polymerase sequences (of 417 nt in length). The 166 sequences were extracted from GenBank (the accession numbers are presented in Appendix A). (**b**) The phylogenetic tree constructed using the central segment of DNA polymerase sequences of MastAdV genes extracted from GenBank (the accession numbers are presented in Appendix A). The numbers in the branches indicate the bootstrap values. Most appropriate substitution model according to BIC: TIM 3+F+I+G4. The novel MastAdV BaAdV/MOW15-Nn19/Quixote is marked in bold and purple; its closest relatives in the clade are only in purple (all from the Vespertilionidae bat). The novel MastAdV BatAdV/MOW15-Pn21 is marked in bold and blue; other MastAdVs from the Vespertilionidae bat are marked only in blue. The MastAdVs whose hosts belong to other bat families are highlighted in brown.

**Figure 4 viruses-16-01207-f004:**
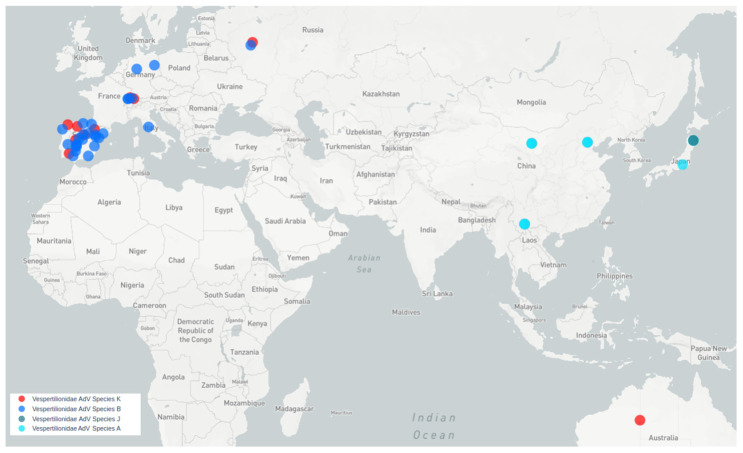
Map of MastAdVs detected in Old World and Australasian Vespertilionidae bats. Blue circles indicate viruses of species B. Red circles indicate viruses of a presumably new bat mastadenovirus species, tentatively named K (genetically similar to Quixote). Aqua circles—species A, dark aqua circles—species J.

**Table 1 viruses-16-01207-t001:** Characterization for all of the newly predicted genes found in the BatAdV/MOW15-Nn19 (PP297886) and their most similar genes from the NCBI database. D = DNA synthesis and packaging; H = host defense subversion (putative); R = regulation, S = structural. Genes and proteins of unknown function and/or homology are shaded gray. Asterisks (*) = proteins encoded by isolate WA3301 (acc. num. MK472072.1).

	Gene Coordinates, Encoding Protein Length	Gene Name	Protein Description; Function, Accordingly to ICTV	Transcription Class	Nearest Homolog (Proteins)		
					Genbank ID	Identity	Query Coverage, %
1	372–968, 199 aa	*E1A*	R, E1A; Modulation of the host cell transcriptional machinery	E1	QGX41997 *	39.13%	74%
2	1034–1258, 75 aa	*ORF2*	Putative protein		Unknown	-	-
3	1050–1289 complement, 80 aa	*ORF33*	Putative protein		Unknown	-	-
4	1422–2150, 243 aa	*E1B 19K*	R, E1B small; Modulation of the host cell transcriptional machinery		QGX41989 *	39.92%	96%
5	1811–3271, 487 aa	*E1B 55K*	R, E1B Large; Modulation of the host cell transcriptional machinery		QGX41980 *	47.00%	99%
6	3271–3621, 117 aa	*ORF32*	Putative protein	Intermediate	Unknown	-	-
7	3602–4270, 223 aa	*IX*	S (IX); Capsid minor protein		QGX41995 *	53.55%	66%
8	4274–5344 complement, 357 aa	*IVa2*	D, S (IVa2); Capsid minor protein		QGX41983 *	86.52%	99%
9	5371–8619 complement, 1083 aa	*POL*	D; DNA polymerase	E2b	QGX41975 *	76.90%	98%
10	8619–10,250 complement, 544 aa	*pTP*	D; pTP, Preterminal protein; Important role in the initiation of viral DNA replication		QGX41978 *	83.39%	99%
11	10,530–11,684, 445 aa	*L1_1*	D; pP3 52k	L1	QGX41986 *	73.42%	99%
12	11,629–13,278, 550 aa	*L1_2*	D, S (pIIIa); Pre-hexon-linking protein, phosphoprotein Capsid minor protein		QGX41979 *	80.00%	99%
13	13,382–14,776, 465 aa	*L2_1*	S (III); penton base; Capsid major protein	L2	QGX41981 *	84.70%	99%
14	14,810–15,349, 180 aa	*L2_2*	S (pVII), major core; Associated with the DNA and form the core within the virion		QGX41998 *	74.51%	27%
15	15,412–16,509, 366 aa	*L2_3*	S (V), minor core; Associated with the DNA and form the core within the virion		QGX41987 *	70.89%	98%
16	16,455–16,754 complement, 100 aa	*ORF14*	Putative protein		Unknown	-	-
17	16,585–16,821, 79 aa	*L2_4*	S (pX), pre-core protein X; Associated with the DNA and form the core within the virion		YP_009388318	61.19%	84%
18	16,908–17,657, 250 aa	*L3_1*	S (pVI); Capsid minor protein	L3	QGX41988 *	72.31%	99%
19	17,767–20,466, 900 aa	*L3_2*	S (II), hexon; Capsid major protein		QGX41976 *	83.70%	99%
20	20,468–21,088, 207 aa	*L3_3*	D, S, protease; Peptidase_C5		QGX41993 *	79.41%	98%
21	21,128–22,564 complement, 479 aa	*DBP*	D, DBP; DNA binding protein	E2a	QGX41982 *	58.44%	99%
22	22,594–24,696, 701 aa	*L4_1*	D, 100 kDa Shutoff	L4	QGX41977 *	72.38%	92%
23	24,542–25,042, 157 aa	*L4_2*	D, R, phosphoprotein 2, pP2		QGX41999 *	56.73%	56%
24	25,332–26,099, 256 aa	*L4_3*	S (pVIII), hexon associated protein; Capsid minor protein		QGX41992 *	85.15%	89%
25	26,083–26,478, 132 aa	*E3S*	H, 14,4 kDA protein	E3	YP_010796290	30.65%	80%
26	26,533–30,003, 1157 aa	*E3L*	H, pE3L		QGX41974 *	29.21%	99%
27	30,059–32,377, 773 aa	*pIV*	S (IV), fiber; Capsid major protein	L5	QGX41984 *	33.24%	56%
28	32,416–32,874 complement, 153 aa	*E4 Orf6/7*	R, E4 protein; Modulation of the host cell transcriptional machinery	E4	AGT77890	44.59%	36%
29	32,899–33,717 complement, 273 aa	*E4 Orf6*	R, E4 protein; Modulation of the host cell transcriptional machinery		QGX41990 *	36.69%	89%
30	33,687–34,226, 180 aa	*E4 Orf26*	Putative protein		Unknown	-	-
31	34,207–34,572, 122 aa	*E4 Orf27*	Putative protein		Unknown	-	-
32	34,584–35,285, 234 aa	*E4 Orf28*	R, Putative dUTPase		QGX41994 *	27.07%	75%
33	35,313–36,422, 370 aa	*E4 Orf29*	ORF19, Putative protein		QGX41985 *	35.94%	86%
34	36,476–37,060, 195 aa	*E4 Orf30*	R, Putative dUTPase		QGX41991 *	55.61%	95%
35	37,094–37,558, 155 aa	*E4 Orf31*	Putative protein		Unknown	-	-

## Data Availability

The raw sequencing data are deposited in the NCBI SRA under the accession numbers SRX11823236 and SRX11824039. The genome sequences of BatAdV/MOW15- Nn19 and BatAdV/MOW15-Pn21 have been deposited in the GenBank under accession numbers PP297886 and PP386572-PP386582.

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
