# Peer review of "A Novel Mastadenovirus from Nyctalus noctula which Represents a Distinct Evolutionary Branch of Viruses from Bats in Europe"

_viruses, 2024, doi:10.3390/v16081207_

Round 1

Reviewer 1 Report

Comments and Suggestions for Authors

The manuscript by Speranskaya et al. provides interesting data on the metagenomic identification of novel mastadenovirus from bats of Eastern Europe.

The manuscript deserves to be published but only after major revisions, as the text of the manuscript needs to be significantly improved.

1. L. 2. Please, write the species name of the bat in the title of the manuscript in full.

2. L. 3. The phrase "Viruses Infecting Bats" in the title is debatable. Multiple studies show that bats are asymptomatic carriers/hosts of a large abundance of viruses. Please, consider rewriting the title.

3. L. 5-10. Lomonosov Moscow State University is mentioned in the affiliations, but none of the authors is affiliated with this university.

4. L. 11-34. The abstract is not written according to the authors' guidelines. The maximum amount of words should be limited to 200 words. The use of headings should be avoided.

5. The whole manuscript. The first mention of species names in the abstract and the text of the manuscript should be in full and written in italics. Then it is acceptable to use genus shorthand while mentioning the species name a second time or more. The manuscript should be revised according to these generally accepted rules.

6. The whole manuscript. Please, consider italicizing species, genus, and family of a virus when used in a taxonomic sense.

7. The whole manuscript. Some sentences miss periods at the end and have extra spaces. This should be revised.

8. L. 57. Please, confirm that "Canine adenoviruses" need to be written in italics.

9. L. 104-123. Please, mention the manufacturer and country of origin of each kit/equipment used.

10. L. 124-145. Were the contaminations of the raw reads checked by aligning them to human/bat genomes?

11. L. 138. The phrase "de-reference" is not widely used. Please, consider rewriting. 

12. L. 163. Please, cite your previous work. It is not clear which work you mention without looking at the author profiles.

13. L. 164. What is the "fam."? Family? Needs to be clarified.

14. L. 182. Please, confirm that the resolution of the figure is 300 dpi or higher. Original images must be at a sufficiently high resolution.

15. L. 276-277 and L. 362. Please, see the comment 2.

16. L. 294-309. The discussion of the prevalence of mastadenoviruses in bat populations in this article might be not relevant as the sample size of bats included in this study (12 individuals) does not allow for proper comparison. I recommend either deleting this paragraph or highlighting the limitations of this study (e.g. sample size) at the end of this paragraph, as the text in its current form could be confusing to the general audience.

17. L. 310-334. The text in this paragraph doubles the text provided in the Introduction. Please, either delete this text from the Introduction/Discussion or rewrite it.

18. L. 345-348. I suggest authors expand the discussion of the compliance of the discovered virus with the criteria for recognition as a new species.

19. L. 376-388. The methodology of the generation of the figure and the software used for this need to be described in the M&M section. The figure and its description need to be presented in the corresponding Results section. 

20. L. 481. There is no need to cite previous work in this section.

21. L. 484. There is no database called "GenBank SRA". Please, consider writing NCBI SRA if you used this database for the deposit of the raw data.

I must acknowledge that the data analysis for this report was conducted on a suitable level, but the text of the manuscript and the description of the results need to be significantly improved before publication.

Author Response

Dear Reviewer. The manuscript has been edited according to your requirements. Attached is a file with detailed answers to your questions.

1. L. 2. Please, write the species name of the bat in the title of the manuscript in full.

Fixed, “…Nyctalus noctula…”

2. L. 3. The phrase "Viruses Infecting Bats" in the title is debatable. Multiple studies show that bats are asymptomatic carriers/hosts of a large abundance of viruses. Please, consider rewriting the title.

Fixed, “…Viruses from Bats in Europe…”

3. L. 5-10. Lomonosov Moscow State University is mentioned in the affiliations, but none of the authors is affiliated with this university.

Thank you for your comment - we have corrected the author affiliations to relevant ones.

Kruskop S.V.  Lomonosov Moscow State University

4. L. 11-34. The abstract is not written according to the authors' guidelines. The maximum amount of words should be limited to 200 words. The use of headings should be avoided.

Presently abstract consist of 198 words.

5. The whole manuscript. The first mention of species names in the abstract and the text of the manuscript should be in full and written in italics. Then it is acceptable to use genus shorthand while mentioning the species name a second time or more. The manuscript should be revised according to these generally accepted rules.

Scientific names have been cheked.

6. The whole manuscript. Please, consider italicizing species, genus, and family of a virus when used in a taxonomic sense.

Thank you for your feedback. We have carefully reviewed the entire article and made the necessary corrections.

7. The whole manuscript. Some sentences miss periods at the end and have extra spaces. This should be revised.

Double checking and correction of typing errors and missing characters has been done.

8. L. 57. Please, confirm that "Canine adenoviruses" need to be written in italics.

It's fixed

9. L. 104-123. Please, mention the manufacturer and country of origin of each kit/equipment used.

Thank you for this remark, we have added the missing names of the manufacturing countries.

  1. QIAamp Viral RNA Mini Kit (Qiagen, Germany)

  2. Reverta-L (AmpliSens, Russia)

  3. NEBNext Ultra II Non-Directional RNA Second Strand Module (E6111L, New England Biolabs, USA)

  4. H2O (milliQ, Merck Millipore, Germany)

  5. NEBNext Ultra II DNA Library Prep Kit for Illumina (New England Biolabs, USA)

  6. Nextera XT Index Kit (Illumina, USA)

  7. Illumina HiSeq 1500 (Illumina, USA)

  8. HiSeq PE Rapid Cluster Kit v2 and HiSeq Rapid SBS Kit v2 (500 cycles) (Illumina, USA).

10. L. 124-145. Were the contaminations of the raw reads checked by aligning them to human/bat genomes?

Preliminary analysis showed that there were almost no human reads in the metageomic sequences, so there was no need to filter them out. Pipistrellus nathusii reads were also not filtered out because at a time when all of the bioinformatics analysis was performed, there was no available Pipistrellus nathusii genome available.

11. L. 138. The phrase "de-reference" is not widely used. Please, consider rewriting. 

This sentence wes rewritten

“…Genome annotation was performed using the following software: de novo using Prokka software (Seemann, 2014) and reference based using MK472072.1 as reference, GATU …”

12. L. 163. Please, cite your previous work. It is not clear which work you mention without looking at the author profiles.

Thanks for the comments - we've fixed it.

“… (Speranskaya et al, 2023)…”

13. L. 164. What is the "fam."? Family? Needs to be clarified.

“…family…”

14. L. 182. Please, confirm that the resolution of the figure is 300 dpi or higher. Original images must be at a sufficiently high resolution.

Confirmed

15. L. 276-277 and L. 362. Please, see the comment 2.

Thanks for the comments - we've fixed it

“…viruses from bats…”

16. L. 294-309. The discussion of the prevalence of mastadenoviruses in bat populations in this article might be not relevant as the sample size of bats included in this study (12 individuals) does not allow for proper comparison. I recommend either deleting this paragraph or highlighting the limitations of this study (e.g. sample size) at the end of this paragraph, as the text in its current form could be confusing to the general audience.

Thanks for this comment, we agree that our sample size is not quite sufficient for such conclusions.

17. L. 310-334. The text in this paragraph doubles the text provided in the Introduction. Please, either delete this text from the Introduction/Discussion or rewrite it.

We've tweaked the discussion, partially moving the information to the introduction. And completely delete lines 310-334.

18. L. 345-348. I suggest authors expand the discussion of the compliance of the discovered virus with the criteria for recognition as a new species.

Discussin was revised

19. L. 376-388. The methodology of the generation of the figure and the software used for this need to be described in the M&M section. The figure and its description need to be presented in the corresponding Results section. 

Maps were generated using the Python (v3.9.17) Plotly package (v5.9.0) [1, 2]. Briefly, the corresponding sample coordinates were transformed to latitude and longitude, stored as a table with respect to the sample metadata, and plotted from the dataframe using Plotly Express.

[1] Van Rossum, G., & Drake, F. L. (2009). Python 3 Reference Manual. Scotts Valley, CA: CreateSpace.

[2] Plotly Technologies Inc. Collaborative data science. Montréal, QC, 2015. https://plot.ly.

20. L. 481. There is no need to cite previous work in this section.

Thanks. Corrected.

21. L. 484. There is no database called "GenBank SRA". Please, consider writing NCBI SRA if you used this database for the deposit of the raw data.

Done

Reviewer 2 Report

Comments and Suggestions for Authors

Speranskaya et al. present an interesting report on the characterization of a novel Bat Mastadenovirus that could potentially represent the first complete genome of a representative of a novel species among the Bat mastadenoviruses. Despite the relevance of these results, the manuscript has multiple issues that require addressing before the report can be accepted. Following my concerns and suggestions:

1. Is the abstract structured? Maybe you don't need to explicitly write the words 'Aims', 'Methods and Results' and 'Conclusions.

2. After reading the whole manuscript I understand why 'K' is assigned as the possible new species, but most readers will be wondering why 'K'. Maybe just adding a line establishing that "K, as the next available letter for the species, was provisionally assigned as the species of this sample"

3. The whole first paragraph of the introduction is unnecessary as it is a lot of description about human mastadenovirus, but that's not necessary for the main topic of the report. It can be summarized as simple as "some mastadenovirus infect humans too". Notice how the first sentence of the second paragraph already makes the first paragraph unnecessary: "MastAdVs infect a variety of mammalian hosts" (as humans are mammals, we fall in this group).

4. The second paragraph of the introduction is a bit contradictory. "in general, MastAdvs appear to be host-specific viruses", however the rest of the sentences are examples of host-switching events and Canine adenoviruses infect multiple host species. Then, just state that although adenoviruses are frequently characterized as host-specific, the multiple examples of host-switching adenoviruses suggest some broader host range for some of them.

5. Materials and methods section looks well described with a single exception that becomes more crucial when reading the results. What is the criterion for the consensus sequence? How much was the depth of the coverage for the assembled consensus? Also, are gapped section of the alignments being removed? Finally, the methods seem to lack a description for the partial sequences analysis of shorter sequences matching your two assembled genomes (partial and complete); what inclusion criteria was used? how many sequences were considered? what date was GenBank accessed to obtain these sequences?

6. In Results, why was total RNA sequencing data used for detection of adenovirus? Adenovirus is double stranded DNA. I have some guesses, but the reason should be stated here. Also, if you will refer to your previous work, include the citation.

7. Is it normal for Bat adenovirus to have names? I liked the association with "Pansa"; however, it seems cumbersome dragging the name and denomination across the manuscript. My suggestion is if you will use the name or the denomination, keep it consistent across the manuscript to avoid confusion in the readership.

8. Similar to the comment above, for the Australian bat adenovirus sequence, choose a name, introduce it once and then use it consistently across figures and text. In some points the name is the denomination, others as Australian and others as WA3301 and others as the accession number. This gets difficult to follow for a reader who is not an expert in bat adenoviruses.

9. In the first sentence of the section 3.2, I noticed that you assured the inclusion of the ITR regions, could you comment on how it was confirmed that the terminal regions were fully sequenced?

10. In figure 1, the description talks about Dark-Red arrows and omits the description of the yellow/golden arrows. Maybe this needs correction.

11. Line 197, what is 're-core'?

12. Identity at 44.59% seems already low identity. Also, percentages without information on the length of the compared segments, proteins or sequences can be misleading, I suggest to add the length in amino acids or nucleotides.

13. In section 3.2, the fourth paragraph should go before the third paragraph, as usually the description would present the evidence and then conclude that there is enough evidence under the ICTV criteria to consider a new species. Furthermore, the section 3.2 maybe should end with the current fourth paragraph.

14. In line 216-218 states that other species closest to the new virus had shorter lengths; are those complete genomes? Could the accession numbers be included for reference between parentheses?

15. The fifth paragraph of section 3.2 is unnecessary or incomplete, as establishes the E3 region of Quixote and WA33-1 are the same but both differ in the E4 region. Could you explain how they are same and how they differ? More ORF's, high identity, low identity?

16. In the third paragraph of section 3.2, what is "0.53% total length? This represents approximately 200nt. Similarly '0.27 total length' could mean 10237 nt.

17. Figure 2 is missing a scale making impossible to read whether the branch lengths are showing big or short distances.

18. What is "identifien" in line 251?

19. The partial sequence of BatAdV/MOW15-Pn21 is stated to be 7112 bp formed from 11 contigs, but what sections of the genome or how long are those contigs is not stated (maybe a diagram could make this sequence less abstract and more reliable for further analysis).

20. Although the approach aligning the DNA polymerase of multiple samples to prove the relation of other viruses to the sequences accomplished in this study, more information (methods) should be provided for example the number of sites without gaps considered for the tree and the average identity of those other sequences to the sequences of this study.

21. Figure 3a is also lacking a scale to make the tree interpretable. Without a scale the current tree is just a cladogram.

22. In the description of Figure 3 it just states terminal part of the DNA polymerase, the 5' or 3' terminal part? 100nt, 1000nt or 3000nt of that terminal part?

23. The discussion starts with 'Adenoviruses have been in evidence in bats across Europe'; however I can't understand the intention of that statement, do you mean evidence of infection? transmission? establishing active infections? establishing active infections?

24. Section 4.2 should be edited to be more concise.

25. Section 4.3 is too contradictory, hypothetical and overall unnecessary. It is a long divagation on how bats got from Asia to Australia, when maybe this is not relevant for the present study, to finally suggest that other intermediary bats could be the method for transmission. Also, missing the point that maybe other uncharacterized non-bat hosts could be a common origin for infections in Europe and Australia. There too many assumptions and "presumably" to be considered a logical deduction. With enough genetic data for hosts and pathogens a more informed analysis can be performed to assess the hypothesis of host restriction. In any case, as presented in the results, the closest virus is already diverged enough to be considered from a separate species.

26. Figure 4 should be better explained or removed. Blue dots are not 'K' bats, but in different tones and red are similar to 'K' (how much similarity? what criteria?).

27. Section 4.4 seems to be describing what lacks on the results about the similarities and differences for E3 and E4, however, as there is a lack of functional characterization for these ORFs, assuming that they are necessary genes in mastadenovirus is a bit overestimated.

28. In the second paragraph of section 4.4, the coordinates of E1A and Pol indicating Val rather than Met can be explained by artifact of the sequencing (without the sequence depth, I am more inclined to believe the sequence quality resolved a 'G' where it was an 'A'). Alternatively, polymerase can be initiated from a separate exon in many adenovirus species.

29. In line 389 what is n.a.

Comments on the Quality of English Language

The punctuation at the end of sentences needs to be revised after citations.

Line 74: ) .

Line 197: re-core

Line 327-328: uncleare

Author Response

Dear Reviewer. The manuscript has been edited according to your requirements. Attached is a file with detailed answers to your questions.

1. Is the abstract structured? Maybe you don't need to explicitly write the words 'Aims', 'Methods and Results' and 'Conclusions.

We have reformatted the abstract to make it more readable.

2. After reading the whole manuscript I understand why 'K' is assigned as the possible new species, but most readers will be wondering why 'K'. Maybe just adding a line establishing that "K, as the next available letter for the species, was provisionally assigned as the species of this sample"

Thanks for the recommendation, that's what we did, we added this sentence to the abstract.

3. The whole first paragraph of the introduction is unnecessary as it is a lot of description about human mastadenovirus, but that's not necessary for the main topic of the report. It can be summarized as simple as "some mastadenovirus infect humans too". Notice how the first sentence of the second paragraph already makes the first paragraph unnecessary: "MastAdVs infect a variety of mammalian hosts" (as humans are mammals, we fall in this group).

We have shortened the information on human mastadenoviruses in the introduction. However, some information on human mastadenoviruses was left. The reason is simple: new viruses in wild mammals are studied primarily to assess the risk of transmission to other mammals, including humans.

4. The second paragraph of the introduction is a bit contradictory. "in general, MastAdvs appear to be host-specific viruses", however the rest of the sentences are examples of host-switching events and Canine adenoviruses infect multiple host species. Then, just state that although adenoviruses are frequently characterized as host-specific, the multiple examples of host-switching adenoviruses suggest some broader host range for some of them.

No, there is no contradiction. IN GENERAL, MastAdVs appear to be host-specific viruses, but there are EXCEPTIONS

We understand your concern about the potential contradiction. But our intention was to emphasize that although in the scientific literature adenoviruses, including mastadenoviruses, are usually described as host-specific viruses, there are notable exceptions that demonstrate cases of host switching.

Actually, it is the exceptions to the rules that make us look for the following "exceptions to the rules", In our opinion, this is not strict host specificity that is crucial for this and further research, since the classification of these viruses as primarily specific to certain hosts does not necessarily determine their safety for people and animals in close contact with people.

We have made adjustments to this paragraph to make it clearer and less controversial.

5. Materials and methods section looks well described with a single exception that becomes more crucial when reading the results. What is the criterion for the consensus sequence? How much was the depth of the coverage for the assembled consensus? Also, are gapped section of the alignments being removed? Finally, the methods seem to lack a description for the partial sequences analysis of shorter sequences matching your two assembled genomes (partial and complete); what inclusion criteria was used? how many sequences were considered? what date was GenBank accessed to obtain these sequences?

For BatAdV/MOW15-Nn19/Quixote. The final consensus was 37,915 nt long (average coverage is 19,4, median coverage is ~45). Since ITR sequences at the 5’-and 3’-terminal ends of the genome were identified, we considered the assembled genome to be complete. The coverage of the assembled genome is really good, see attached figure. If necessary, we will include this figure in the supplementary data for publication.

Supplementary data A. Chart for read coverage for assembled genome of BatAdV/MOW15-Nn19/Quixote

6. In Results, why was total RNA sequencing data used for detection of adenovirus? Adenovirus is double stranded DNA. I have some guesses, but the reason should be stated here. Also, if you will refer to your previous work, include the citation.

Thank you for this comment methodological question. The samples were not treated with DNAase, so the DNA remained in the resulting nucleic acid extracts. We added this information in Matereial and Methods.

7. Is it normal for Bat adenovirus to have names? I liked the association with "Pansa"; however, it seems cumbersome dragging the name and denomination across the manuscript. My suggestion is if you will use the name or the denomination, keep it consistent across the manuscript to avoid confusion in the readership.

In general, it's OK to name any virus (not just adenoviruses). Laboratories working with viruses give names to their research objects, and these names take root and become proper names. See, for example, a recent article on the MEMV virus. It's short for Mecsek Mountains virus (https://www.ncbi.nlm.nih.gov/pmc/articles/PMC10250479/). There are other proper names in the same article.

Usually, it's named after the area where the virus was found. But we foundBut we're onBut we have oursBut we foundBut we found the nBut we found ours But we found our virus in mice near Moscow, along with a very large number of other viruses. We had to come up with unique names, otherwise everyone would be confused.

Our virus was called Quixote because we found a second adeno-associated virus in the same bat. We don't describe it, so - yes - the name is not obvious. We've removed the information about "Pansa".

I have also corrected the name of the novel virus throughout the text, it is more correct to call it BatAdV/MOW15-Nn19/Quixote. Quixote for short.

8. Similar to the comment above, for the Australian bat adenovirus sequence, choose a name, introduce it once and then use it consistently across figures and text. In some points the name is the denomination, others as Australian and others as WA3301 and others as the accession number. This gets difficult to follow for a reader who is not an expert in bat adenoviruses.

Thank you for this necessary suggestion, we have unified the name of this virus in the text of the article.

...isolate WA3301...”

9. In the first sentence of the section 3.2, I noticed that you assured the inclusion of the ITR regions, could you comment on how it was confirmed that the terminal regions were fully sequenced?

Thank you for this question, in response to it we consider it possible to give the following explanations:

1) The inverted terminal repeats of ITR with a length of 59 bp, the very presence of which is characteristic of the genomes of mastadenoviruses, have a length relevant to the literature in the range of 35-368 bp (Benko et al., 2022).

2) The sample preparation method for sequencing (DNA fragmentation and subsequent ligation of adapters) was supposed to result in ligation of adapters to 3- and 5- ends of genomic DNA molecules. The 3- and 5- ends of consensus sequences covered by high reads number. Of course, all this is only circumstantial evidence.

3) In our article we only state that we have found terminal inverted repeats of a certain length;

10. In figure 1, the description talks about Dark-Red arrows and omits the description of the yellow/golden arrows. Maybe this needs correction.

Description is corrected.

Yellow arrows indicate genes that encode proteins that modulate the host cell's transcriptional machinery.

11. Line 197, what is 're-core'?

Fixed, “… pre-core…”

12. Identity at 44.59% seems already low identity. Also, percentages without information on the length of the compared segments, proteins or sequences can be misleading, I suggest to add the length in amino acids or nucleotides.

To clarify, we add an additional link to Table 1. This table includes not only the identity percentage of 44.59% but also indicates a coverage of 36% and amino acid length of proteins encoded by genes including E4 Orf6/7.

13. In section 3.2, the fourth paragraph should go before the third paragraph, as usually the description would present the evidence and then conclude that there is enough evidence under the ICTV criteria to consider a new species. Furthermore, the section 3.2 maybe should end with the current fourth paragraph.

The order of paragraphs is changed.

3.2. §4 3 The genome organisation of the "Quixote" (isolate BatAdV/MOW15-Nn19) was the same as....

3.2. §3 4 The complete genome of "Quixote" was 37,915 bp and.....

14. In line 216-218 states that other species closest to the new virus had shorter lengths; are those complete genomes? Could the accession numbers be included for reference between parentheses?

Yes, it is complete sequences. Accession numbers are added.

15. The fifth paragraph of section 3.2 is unnecessary or incomplete, as establishes the E3 region of Quixote and WA33-1 are the same but both differ in the E4 region. Could you explain how they are same and how they differ? More ORF's, high identity, low identity?

We have highlighted the similarity of the E3 region and the difference in the E4 region precisely because, according to the ITCV criteria (Benko et al., 2022) on “Species demarcation criteria” and “Genome organization and replication,” these regions are especially significant for species separation in mastadenoviruses. We provide detailed information in Table 1, where all Is in the form QGX419..* belong to the WA3301 isolate.

The explanations have been added to the text

https://ictv.global/report/chapter/adenoviridae/adenoviridae/mastadenovirus

16. In the third paragraph of section 3.2, what is "0.53% total length? This represents approximately 200nt. Similarly '0.27 total length' could mean 10237 nt.

We are talking about the entire genome sequence (in nucleotides). his is a low similarity. For comparison, the homology of two different strains of the same "B" species (from bats caught in different European countries 10 years apart, namely BatAdV-2 [NC_015932.1] and ITA/2018/251170-16 [MK625182.1]) is 0.99 (100% coverage). Another example, the homology of representatives of species "B" and species "G" is 0.77 with query coverage 35% (NC_015932.1 vs NC_031948.1).

17. Figure 2 is missing a scale making impossible to read whether the branch lengths are showing big or short distances.

Fixed

18. What is "identifien" in line 251?

Fixed. “…Identified…”

19. The partial sequence of BatAdV/MOW15-Pn21 is stated to be 7112 bp formed from 11 contigs, but what sections of the genome or how long are those contigs is not stated (maybe a diagram could make this sequence less abstract and more reliable for further analysis).

The assembled contigs сoordinates relative to the reference genome of Bat adenovirus 2 [NC_015932.1] see Supplementary B (added)

Supplementary B - MOW15-Pn21_contigs with coordinates.fasta

20. Although the approach aligning the DNA polymerase of multiple samples to prove the relation of other viruses to the sequences accomplished in this study, more information (methods) should be provided for example the number of sites without gaps considered for the tree and the average identity of those other sequences to the sequences of this study.

It is not entirely clear what is meant. Phylogenetic analysis of DNA polymerase, including short stretches of the DNA polymerase gene, is a typical approach used to establish virus relatedness. It is not clear why the average identity of these other sequences with the sequences in this study needs to be described.

21. Figure 3a is also lacking a scale to make the tree interpretable. Without a scale the current tree is just a cladogram.

Fixed

22. In the description of Figure 3 it just states terminal part of the DNA polymerase, the 5' or 3' terminal part? 100nt, 1000nt or 3000nt of that terminal part?

t is 5'-terminal part of the polymerase gene of 417 nt length. It is It's consistent with 5462...5878 (complement) position in genome of BaAdV/MOW15-Nn19/Quixote.

The necessary explanations have been added to the picture caption.

23. The discussion starts with 'Adenoviruses have been in evidence in bats across Europe'; however I can't understand the intention of that statement, do you mean evidence of infection? transmission? establishing active infections? establishing active infections?

Thank you for your inquiry. We would like to clarify a phrase you mentioned, as both in our research and in the research of colleagues cited later in this paragraph, viral sequences have been detected in fecal samples from bats. We simply state that adenoviruses are widespread in European bat populations. The question of whether or not the infection process in bats is painful or asymptomatic has not been addressed by us due to lack of data. Regarding the possibility of transmission of viral infection between bats and other mammals, this is one of the objectives of our study - to provide more detailed information on mastadenoviruses found in bats, which could be useful for future risk assessments of secondary infections and predicting and preventing viral outbreaks.

24. Section 4.2 should be edited to be more concise.

Text was rewritted and to shortened by lines 310-334.

25. Section 4.3 is too contradictory, hypothetical and overall unnecessary. It is a long divagation on how bats got from Asia to Australia, when maybe this is not relevant for the present study, to finally suggest that other intermediary bats could be the method for transmission. Also, missing the point that maybe other uncharacterized non-bat hosts could be a common origin for infections in Europe and Australia. There too many assumptions and "presumably" to be considered a logical deduction. With enough genetic data for hosts and pathogens a more informed analysis can be performed to assess the hypothesis of host restriction. In any case, as presented in the results, the closest virus is already diverged enough to be considered from a separate species.

Text was rewritted. Some of the reasoning we have left out. It seems to us important to discuss exactly the "natural" spread of viruses by "natural" migration of bats. But this issue is rarely detailed in the scientific literature. Usually discussion is limited to general statements.

We have tried to shorten the text and make it more logical and understandable.

26. Figure 4 should be better explained or removed. Blue dots are not 'K' bats, but in different tones and red are similar to 'K' (how much similarity? what criteria?).

Figure 4 was improoved

27. Section 4.4 seems to be describing what lacks on the results about the similarities and differences for E3 and E4, however, as there is a lack of functional characterization for these ORFs, assuming that they are necessary genes in mastadenovirus is a bit overestimated.

Thank you for your thoughtful objection. Text was rewritted

28. In the second paragraph of section 4.4, the coordinates of E1A and Pol indicating Val rather than Met can be explained by artifact of the sequencing (without the sequence depth, I am more inclined to believe the sequence quality resolved a 'G' where it was an 'A'). Alternatively, polymerase can be initiated from a separate exon in many adenovirus species.

Then the coordinates were manually checked using *.bam files with careful consideration using the closest complete genome as a reference (accession number MK472072.1). For E1A gene, the start-codon position was covered by 71-72 reads (GTG sequence nucleotide count - 99-100%). For POL gene, the start-codon coverage 70 reads, GTG (reverse complement of CAC) sequence nucleotide count - 97-100% Some findings suggest that valine tRNA may be involved in the initiation of sgRNA translation (Sanz et al., 2019). Therefore, we suggest possible molecular mechanisms leading to the replacement of Val by Met during translation.

29. In line 389 what is n.a.

The abbreviation "nt" for "nucleotide" is used throughout the text.

Comments on the Quality of English Language

The punctuation at the end of sentences needs to be revised after citations.

Line 74: ) .

Thanks for the comments - we've fixed it.

Line 197: re-core

Fixed, “…pre-core…”

Line 327-328: uncleare

Fixed, “…unclear…”

Round 2

Reviewer 1 Report

Comments and Suggestions for Authors

The authors addressed most of the comments.

However, there are still some issues.

1. "Canine adenoviruses" are written in italics at P. 1, L. 39, and then this term is written in ordinary font at L. 527. The authors should clarify this. According to ICTV "canine adenoviruses" are written in ordinary font: https://ictv.global/report/chapter/adenoviridae/adenoviridae

2.  L. 1058-1059. There is no need to write "...that we cite".

3. L. 1030, 1323. The authors state that the detected mastadenoviruses infect studied bats. Please provide evidence that there were signs of infections in studied bats, or rewrite these sentences.

4. Figure 4 needs to be placed in the Results section, where the authors should describe the results of the visualization of the distribution of mastadenoviruses in Old World and Australasian Vespertilionidae bats. Then in the Discussion section, the authors should discuss these results.

Author Response

  1. "Canine adenoviruses" are written in italics at P. 1, L. 39, and then this term is written in ordinary font at L. 527. The authors should clarify this. According to ICTV "canine adenoviruses" are written in ordinary font: https://ictv.global/report/chapter/adenoviridae/adenoviridae

Fixed

  1. L. 1058-1059. There is no need to write "...that we cite".

Fixed

  1. L. 1030, 1323. The authors state that the detected mastadenoviruses infect studied bats. Please provide evidence that there were signs of infections in studied bats, or rewrite these sentences.

replaced by “circulated”

  1. Figure 4 needs to be placed in the Results section, where the authors should describe the results of the visualization of the distribution of mastadenoviruses in Old World and Australasian Vespertilionidae bats. Then in the Discussion section, the authors should discuss these results.

Fixed

Reviewer 2 Report

Comments and Suggestions for Authors

Thanks to the authors for considering my suggestions and also for the responses to my observations. I consider that all my concerns have been properly addressed.

Author Response

>Thanks to the authors for considering my suggestions and also for the responses to my observations. I consider that all my concerns have been properly addressed.

Thanks